# Influence of Algae Supplementation on the Concentration of Glutathione and the Activity of Glutathione Enzymes in the Mice Liver and Kidney

**DOI:** 10.3390/nu13061996

**Published:** 2021-06-10

**Authors:** Grażyna Świderska-Kołacz, Małgorzata Jefimow, Jolanta Klusek, Norbert Rączka, Szymon Zmorzyński, Anna Wojciechowska, Iwona Stanisławska, Marek Łyp, Joanna Czerwik-Marcinkowska

**Affiliations:** 1Institute of Biology, Jan Kochanowski University, 25-420 Kielce, Poland; grazyna.swiderska-kolacz@ujk.edu.pl (G.Ś.-K.); jolanta.klusek@ujk.edu.pl (J.K.); norbert.raczka@ujk.edu.pl (N.R.); 2Department of Animal Physiology and Neurobiology, Nicolaus Copernicus University, 87-100 Toruń, Poland; jefimow@umk.pl; 3Department of Cancer Genetics and Cytogenetics Laboratory, Medical University, 20-080 Lublin, Poland; szymon.zmorzynski@umlub.pl; 4Department of Geobotany and Landscape Planning, Nicolaus Copernicus University, 87-100 Toruń, Poland; ankawoj@umk.pl; 5College of Rehabilitation, 01-234 Warszawa, Poland; iwona.stanislawska@cr.edu.pl (I.S.); marek.lyp@cr.edu.pl (M.Ł.)

**Keywords:** algae, EPA, supplementation, glutathione, glutathione enzymes, oxidative stress

## Abstract

Algae are potential and natural source of long-chain polyunsaturated fatty acids like eicosapentaenoic acid (EPA) and docosahexaenoic acid (DHA). The diatom *Pinnularia borealis* accumulates high levels of EPA and may be considered as a source for commercial production of dietary supplements. In this study we asked the question whether diet supplementation with *P. borealis* may augment antioxidant defense and ameliorate risk factors for cardiovascular diseases. We fed mice (*Mus musculus*) with lyophilized diatom solutions of different concentrations (1%, 3%, and 5%) for 7 days. Then we measured glutathione content and the activity of glutathione redox system enzymes, total cholesterol and triacylglycerol concentrations, and malondialdehyde concentration in the liver and kidney. We found that cholesterol and triacylglycerol concentrations in the liver and kidneys were the lowest in mice who were fed with the highest concentration of *Pinnularia borealis,* suggesting protective properties of algae. Additionally, the lowest concentration of *Pinnularia borealis* was sufficient to improve antioxidant capacity. Our results suggest that *P. borealis* may be used as a source for dietary supplements rich in EPA, but the amount supplied to the organism should be limited.

## 1. Introduction

Free radicals and ROS pose significant challenges to the organisms as they may damage nucleic acids, proteins and lipids. Thus, organisms have evolved effective antioxidant defense system which includes endogenous enzymatic and nonenzymatic antioxidants. However, antioxidants like vitamin C, vitamin E, carotenoids, and phenols may be also obtained from diet. In addition, polyunsaturated fatty acids (PUFAs), which rapidly oxidize in vitro, may decrease oxidative damage in vivo [1].

A diet rich in PUFAs decreases the risk of cardiovascular disease and some types of cancer. It may also decrease plasma cholesterol concentration and thus attenuates the risk of cardiovascular and Alzheimer diseases [1,2], and plays a beneficial role in psychiatric disorders, vision, asthma, and rheumatoid arthritis [3]. The human body can synthesize many types of fatty acids, but essential fatty acids, such as α-linolenic acid (omega-3, ω-3) and linoleic acid (ω-6), must be obtained from the diet. Two derivatives of α-linolenic acid, the eicosapentaenoic acid (ω-3, EPA) and docosahexaenoic acid (ω-3, DHA), are important structural components of the cell membrane [4]. EPA and DHA ameliorate risk factors for cardiac diseases, such as stroke, arrhythmia, and high blood pressure [3], and play a beneficial role in psychiatric disorders, vision, asthma, and rheumatoid arthritis [4]. Both EPA and DHA are commonly obtained from the daily diet, mainly from fish oil. However, problems such as the odor and instability of extraction products are encountered during extraction from fish [4]. Thus, alternative sources are demanded. Diatoms represent one major group of photosynthetic algae [5,6,7,8,9,10,11] and they are known for high levels of the essential ω-3 fatty acids, EPA, DHA, and carotenoids [12,13]. In 2011, Lang et al. published fatty acid profiles of more than 2000 microalgae, allowing a detailed study of VLC-PUFA profiles in diatoms [14]. They are rich in medium-chain and very long-chain polyunsaturated fatty acids (VLC-PUFAs). The predominant fatty acids are myristic acid (14:0), palmitic acid (16:0), palmitoleic acid (16:1), and eicosapentaenoic acid (20:5), while the C18 fatty acids are usually present in trace amounts [15]. Diatoms are ecologically widespread, occur in marine, freshwater, and terrestrial habitats worldwide, and significantly contribute to primary production and to the global cycling of both carbon and nutrients [16]. *Pinnularia borealis* is a common terrestrial cosmopolitan species found in aerial habitats (rocks, walls, soil, moss) and flowing and standing waters [17]. In recent years, diatoms are increasingly being recognized for their large potential in biotechnological applications, as they produce high-value compounds that can be used in medicine and as food supplements [18].

In this study, we asked the question whether dietary supplementation with diatom *Pinnularia borealis* may augment antioxidant defense in mouse *Mus musculus*. We analyzed the glutathione antioxidant system, which plays crucial role in maintaining oxidative homeostasis, as well as the concentration of malondialdehyde, a product of lipid peroxidation, as a marker of oxidative stress. The glutathione antioxidant system consists of reduced and oxidized forms of glutathione (GSH and GSSH, respectively), glutathione peroxidases, and glutathione reductase. Increased GSSG (glutathione disulfide)-to-GSH ratio is indicative of greater oxidative stress [1]. Glutathione is synthesized in the cytoplasm of the cells and the liver is the most active organ [1]. Reduced glutathione (L-gamma-glutamyl-L-cysteinyl-glycine, GSH) is the most abundant low-molecular-weight thiol in mammalian cells with potent antioxidant properties [19,20,21]. GSH also regenerates other oxidized antioxidants, such as vitamin C and vitamin E. It also takes a part in the repair of the structure of protein, lipid, and nucleic acid molecules damaged by oxidation [22]. Glutathione peroxidases (GPx), with their high affinity for hydrogen peroxide, use the reduced form of glutathione to reduce hydrogen peroxide to water and eliminate it from the body [1]. The GSH redox cycle is closed by glutathione reductase, which catalyzes the reduction of oxidized form of glutathione to its reduced form [1]. This process is important for maintaining proper cellular ratio of glutathione in its reduced and oxidized forms [23]. 

Finally, the glutathione S-transferase (GST) catalyzes the conjugation of GSH to a wide variety of endogenous and exogenous electrophilic compounds [24,25,26]. The GST conjugation is the first step in the mercapturic acid pathway that leads to the elimination of toxic compounds. GST plays a significant role in detoxification of different harmful xenobiotics, both exogenous and endogenous, and in carcinogenic biotransformation processes as well [27]. In mammals, GST is present in all organs and tissues, but the highest content of the enzyme is found in the liver [21]. 

Another goal of our study was to test the prediction that short-term diet supplementation with *Pinnularia borealis* may ameliorate risk factors for cardiovascular diseases. To do this, we measured triacylglycerols and total cholesterol concentrations in the liver and kidneys of laboratory mice.

## 2. Materials and Methods

This study was done at the Jan Kochanowski University in Kielce, Poland. All experimental procedures were approved by the Local Committee for Ethics in Animal Research in Warsaw, Poland (decision number 37/2019). The experiment included two stages of studies: (1) preparation of *Pinnularia borealis* for the use as food supplement and (2) research *in vivo* on 6-week-old Swiss mice.

### 2.1. Preparation of Pinnularia borealis

Algal patches were scraped from pine twigs collected from the Głowoniowa Nyża Cave (Tatra Mountains, Poland), and were placed into sterile plastic bags and transported to the lab, where the algae were cultured on glass Petri plates containing fresh agar made up of 1% Bold’s Basal Medium [28]. The cultures were maintained at a temperature of 20 °C in a 12-h light/12-h dark cycle at 3000 μEm^−2^ s^−1^ l× (40 W cool fluorescent tubes). After approximately two to three months from the first appearance of algae, a microscopic study was performed. The material was observed in living state and identified using a Nikon Eclipse E 600 light microscope (Nikon, Tokyo, Japan), and photographs were taken with a Coolpix 4500 digital camera (Nikon, Tokyo, Japan). The cells were fixed in 3% glutaraldehyde in a 0.1 M phosphate buffer at pH 7.1, post-osmicated in 1% osmium tetroxide, and were then infiltrated in Spurr’s medium [29]; microphotographs were taken with a TESLA BS 500 transmission electron microscope (TEM). *Pinnularia borealis* was identified according to [30] and with an inverted Nikon Eclipse Ti microscope and photographed with Nikon A1 confocal equipment (Nikon, Tokyo, Japan). The 405.488.561 nm laser heads and detection system were a PMT-DU4 detector in the range of 400–820 nm, Plan Apo VC 100XOil inverse lens, Nikon C-HgFiE mercury illumination, and a DS0Fi1C-U3 digital camera. Data acquisition in Nikon NIS-Elements obtained digital images, which were processed with Adobe Photoshop CS5 (Adobe, San Jose, CA, USA).

The fatty acid methyl esters (FAME) profile in dried *Pinnularia borealis* was analyzed by gas chromatography–mass spectrometry (GC-MS) using the Clarus 600T instrument from PerkinElmer (Waltham, MA, USA) based on the method by Guzman et al. [31]. The freeze-dried *P. borealis* content in the ω*n-3* PUFA was as follows: ω*n-3* PUFA (in % of total fatty acids), total = 28%, and EPA = 32%. A total of 14 fatty acids were identified and measured: myristic acid (C14:0), palmitic acid (C16:0), palmitoleic acid (C16:1 *n*-7), hexadecadienoic acid (C16:2 *n*-4), hexadecatrienoic acid (C16:3), stearic acid (C18:0), oleic acid (C18:1 *n*-9), linoleic acid (C18:2 *n*-6), α-linolenic acid (C18:3 *n*-3), arachidonic (C20:4 *n*-6), eicosapentaenoic acid (C20:5 *n*-3), erucic acid (C22:1 *n*-9), and docosahexaenoic acid (C22:6 *n*-3).

### 2.2. Animal Test Subjects and Tissue Processing

#### 2.2.1. Animal Groups and Treatments

A total of 40 male Swiss mice aged 6 weeks with an initial body weight of 20.5 g ± 0.5 g were purchased from the Institute of Genetics and Animal Biotechnology of the Polish Academy of Sciences (Jastrzębiec, Poland). The animals were kept under standard laboratory conditions at a constant temperature of 20–22 °C and a relative humidity of 55% ± 10% in a 12-h light cycle (12L-12D). All individuals were placed in standard, polycarbonate cages (252 × 167 × 140 mm) covered with stainless lids with built-in feed hooper. Food (Field station Łomna-Las, Poland) and water was available ed libitum. Five individuals were placed in one cage. All mice were given a standard food containing (on a per weight basis) 20.5% proteins, 10% fats (0.28 g of PUFA), 62% carbohydrates, 6.5% minerals (calcium 1.9%, phosphorus 1.6%, magnesium 1.2%, sodium 0.9%, and potassium 0.9%), and 1% vitamins (retinol 25.000 IU, cholecalciferol 1.000 IU, thiamin 300 mg, riboflavin 20 mg, pyridoxine 15 mg, cobalamin 40 mg, ascorbic acid 58 mg, quinones 5 mg, tocopherols 125 mg, folic acid 3 mg, biotin 100 µg, nicotinic acid 60 mg, pantothenic acid 35 mg, and choline chloride 1000 mg). The mice had free access to food and water throughout the study. 

After one week of acclimation to laboratory conditions, mice were divided randomly into 4 groups (10 mice per group). For the next 7 days, the mice were fed with different diets: (1) the control group received a standard diet; (2) the second group was fed the standard diet supplemented with 1% solution of lyophilized P. borealis (0.16 mg of EPA); (3) the third group was fed the standard diet supplemented with 3% solution of lyophilized P. borealis (0.48 mg of EPA); and (4) the fourth group was fed the standard diet supplemented with 5% solution of lyophilized P. borealis (0.8 mg of EPA). Lyophilized P. borealis water solutions were given orally once a day in a volume of 50 µL. The mice had free access to food and water throughout the study. The weight gain of the animals was monitored twice (during the experiment), and their food intake was also measured twice. Solutions of lyophilized diatoms were provided with a conventional single-channel p200 micropipette (Gilson Pipetteman, Thermo Fischer Scientific, Reinach, Switzerland) and the pipette tip was offered to the mouth until the mouse began to drink.

#### 2.2.2. Tissue Sample Collection 

After one week of food supplementation, the mice were decapitated and the liver and kidneys were dissected. The liver was perfused with a cooled saline solution (4 °C) to remove any remaining blood. Tissue fragments (100 mg/1 mL buffer) were homogenized with a TH 220-PCRH-Omni TH homogenizer in 0.1 M phosphate buffer with a pH 7.4 containing 10 mM EDTA. Homogenates were centrifuged for 15 minutes at 12.000 rpm in a laboratory centrifuge with cooling MPW-351 R. After centrifugation, the supernatants were immediately frozen in liquid nitrogen and stored at −40 °C for further analysis.

#### 2.2.3. Analytical Methods

In tissue supernatants, the concentration of reduced glutathione, the activity of glutathione peroxidase, glutathione reductase, glutathione transferase, cholesterol, triacylglycerols, protein and malondialdehyde concentrations [32] were measured spectrophotometrically.

### 2.3. Reduced Glutathione (GSH)

The concentration of GSH was determined using Glutathione Assist Kit (Sigma-Aldrich, Darmstadt, Germany, Cat. No. CS0260), according to the manufacturer’s procedure. Samples were first deproteinized with the 5% 5-sulfosalicylic acid solution and centrifuged to remove the precipitated protein. The measurement of GSH uses a kinetic assay in which catalytic amounts of GSH cause a continuous reduction of 5.5 dithiobis (2-nitrobenzoic acid) (DTNB) to 5-thio-2nitrobenzoic acid (TNB), which has a yellow color. The intensity of the color was measured spectrophotometrically at 412 nm.

### 2.4. Glutathione Peroxidase (GPx)

Activity of glutathione peroxidase was measured with Glutathione Peroxidase Cellular Activity Assay Kit (Sigma–Aldrich, Darmstadt, Germany, Cat. No. CPG1). The method is based on the oxidation of glutathione (from GSH to GSSG form) catalyzed by GPx, which is coupled to the recycling of GSSG back to GSH utilizing the glutathione reductase and NADPH. The decrease in NADPH absorbance measured at 340 nm during the oxidation of NADPH to NADP+ is indicative of GPx activity. 

### 2.5. Glutathione Transferase (GST)

Glutathione transferase was measured with a Glutathione S-Transerase (GST) Assay Kit (Sigma–Aldrich, Darmstadt, Germany, Cat. No. CS0410). The assay principle is based on the fact that GST catalyzes the conjugation of L-glutathione to the 1-chloro-2,4-dinitrobenzene (CDNB) through the thiol group of the glutathione. The GS-DNB conjugate absorbs at 340 nm. The rate of increase in the absorption is directly proportional to the GST activity in the sample. 

### 2.6. Glutathione Reductase (GR)

Glutathione reductase activity was measured with a Glutathione Reductase Assay Kit (Sigma–Aldrich, Darmstadt, Germany, Cat. No. GRSA). The activity was measured by the increase in absorbance caused by the reduction of DTNB [5,5”-dithiobis(2-nitrobenoic acid)] at 412 nm. 

### 2.7. Cholesterol Content

Cholesterol content was determined using Biochemtest test by the method of [33]. Cholesterol esters were hydrolyzed by cholesterol esterase (CE) to free cholesterol and fatty acids. Absorption of the test sample was measured at 505 nm.

### 2.8. Triacylglycerols Content

Triacylglycerols content was determined using Alpha Diagnostic test [34]. The assay principle is based on the fact that triacylglycerols are hydrolyzed by lipase to free fatty acids and glycerol, then phosphorylated by ATP. The color intensity was measured at 520 nm.

### 2.9. Malondialdehyde (MDA) Concentration

The MDA concentration was measured spectrophotometrically using the thiobarbituric (TBA) acid assay. To accomplish the assay, 0.5 mL of sample was added to a reaction mixture (1.0 mL) formed by equal parts of 15% tris chloroacetic acid (TCA), in 0.25 *N* HCL, and 0.375% thiobarbituric acid (TBA), in 0.25 *N* HCL, and heated at 95 °C for 30 min. After cooling and centrifugation, the intensity of the pink color of the TBA–malondialdehyde adduct was measured spectrophotometrically at 532 nm [35].

### 2.10. Protein Content

Protein content was determined using method by [36]. The Lowry protein assay is based on the reactions of copper ions with the peptide bonds (with Folin–Ciocalteu reagent, i.e., a mixture of phosphotungstic acid and phosphomolybdic acid) under alkaline conditions (the Biuret test) with the oxidation of aromatic protein residues (mainly tryptophan and tyrosine). Cysteine is also reactive to the reagent. The concentration of the colored complex was measured spectrophotometrically at a wavelength of λ = 750 nm.

### 2.11. Statistical Analysis

All results are presented as means with standard deviations (SD). Biochemical parameters were compared using analysis of variance ANOVA followed by a Tukey post hoc test. Changes in body mass and food consumption were compared using two-sample pair t-test. These analyses were done using the Statistica 9.0 software [37]. The data are also presented as a percentage, assuming control values as 100%. Statistical significance was set at *p <* 0.05 and *p* < 0.01. Furthermore, analyses of direct CCA ordination with Monte Carlo permutation tests were done to determine which of the measured biochemical parameters significantly differentiate the studies sets. These analyses were done using the Canoco 5.0 software [38], separately for the liver and kidneys. 

## 3. Results

The concentration or activity of all parts of glutathione antioxidant system depended on diet supplementation, both in the liver and kidneys. Before the experiment, the mice weighed 20.5 g ± 0.5 g. After one week of supplementation with different diatom solutions, the mice did not differ in body mass [t = −1.75, *p* = 0.088]. Food consumption also did not differ between groups and the mice ate on average 2.5 ± 0.4 g of food per day. However, after one week of supplementation the cholesterol and triacylglycerol contents in the liver and kidneys decreased compared to the control diet, and the activity of glutathione reductase and concentration of malondialdehyde increased significantly, independent of the concentration of diatom solution, in both the liver and the kidneys. 

In the liver, GPx activity decreased continuously with an increase in *Pinnularia borealis* content (F (3, 36) = 3.80, *p* = 0.018) (Figure 1A). The greatest decrease (78.18%) was found in the liver of mice fed with 5% solution (Figure 1A, Table 1). In the kidney, the pattern was almost the opposite (F (3, 36) = 13.08, *p* = 6.2 × 10^−6^) (Figure 1A) and the highest activity of GPx was recorded in animals receiving 3% solution of diatoms (Figure 1A, Table 1). 

Changes in glutathione concentration after dietary supplementation did not show a uniform pattern, either in the liver (F (3, 36) = 5.44, *p* = 0.003) (Figure 1B) or in the kidneys (F (3, 36) = 23.12, *p* =1.6 × 10^−8^) (Figure 1B). After 7 days of food supplementation with the 5% solution, the GSH concentration in the liver decreased to ~82% of the control values but this difference was not significant (Figure 1B, Table 1). A significant difference in GSH content was recorded only between mice fed with the 3% and 5% of diatom solutions. In the kidney, the highest content of GSH was recorded in animals fed with the 3% solution of diatoms (Figure 1B, Table 1). 

The activity of glutathione reductase increased significantly, independent of the concentration of diatom solution, both in the liver (F (3, 36) = 29.34, *p* = 8.9 × 10^−10^) (Figure 1C) and the kidney (F (3, 36) = 9.85, *p* = 7.0 × 10^−5^) (Figure 1C), and in general it was much higher in the liver than in the kidney (*p* < 0.05). GR activity after supplementation with the 5% solution reached up to 290% of the control value in the liver (Figure 1C) and up to 180% of the control value in the kidney (Figure 2, Table 1). 

GST activity in the kidney increased continuously with increasing content of *Pinnularia borealis* in the food (F (3, 36) = 15.59, *p* = 1.2 × 10^−6^) (Figure 1D). The highest activity was recorded after supplementation with the 5% solution, when it reached 158% of the control value (Figure 1D). GST activity in the liver was markedly higher than in the kidney (*p* < 0.05). Although GST activity in liver was dependent on the diet (F (3, 36) = 11.62, *p* = 1.8 × 10^−5^) (Figure 1D), it did not show such a regular pattern as in the kidney (Figure 1D). 

The concentration of malondialdehyde also increased continuously with increasing content of *Pinnularia borealis* in the food, both in the liver (F (3, 36) = 4.26, *p* = 0.011) (Figure 1E) and in the kidney (F (3, 36) = 6.74, *p* = 0.001) (Figure 1E). In both cases, the highest concentration of MDA was recorded after feeding with the 5% solution of diatoms (Figure 1E). 

Variables significant for the diversity of the dataset together accounted for 55.9% of the total variability in liver and 53.9% in the kidney (Figure 2 and Figure 3). In the kidney, a statistically significant relationship between high cholesterol and triglycerides was found in the control samples. The increase in the level of GSH with the 3% supplementation of diatoms was also significant. Similarly, the increase in GR was significantly related to the addition of 5% diatoms (Figure 3). In the liver, the increase in MDA concentration correlated with the 5% admixture of diatoms. The relationship was statistically significant. The increase in the levels of GSH and GR at low concentrations of diatoms in the feed was also statistically significant (Figure 2). 

Triacylglycerols and cholesterol contents in liver and kidney were similar (Figure 1F,G). Diet supplementations with diatoms significantly affected the concentration of triacylglycerols, both in the liver (F (3, 36) = 16.22, *p* = 7.8 × 10^−7^) and in the kidney (F (3, 36) = 7.89, *p* = 0.0004). Comparing to control groups, after the 3% and 5% diatom solutions, the liver triacylglycerols decreased to 75.25% and 65.84%, respectively, and in the kidney to 92 (ns) and 73.16%, respectively. Similarly, diet enrichment with diatoms decreased cholesterol content, both in the liver (F (3, 36) = 2.57, *p* = 0.07) and in the kidney (F (3, 36) = 14.90, *p* =1.8 × 10^−6^). The largest decrease in cholesterol content was recorded in the organs of mice fed with the 5% solution of diatoms. 

## 4. Discussion

In this study we asked the question whether diet supplementation with *Pinnularia borealis* may augment antioxidant defense and ameliorate risk factors for cardiovascular diseases. We found that cholesterol and triacylglycerol concentrations in liver and kidneys were the lowest in mice who were fed with the highest concentration of *P. borealis,* suggesting protective properties of algae. In addition, the lowest concentration of *P. borealis* was sufficient to improve mice antioxidant capacity. 

Algae are potential and natural source of long-chain polyunsaturated fatty acids (PUFAs) like eicosapentaenoic acid (EPA) and docosahexaenoic acid (DHA). In general, lipid content in phototrophic algae ranges from 10% to 60 % of the dry weight, and they contain 10%–20% ω-3 PUFAs [4]. Fatty acid profiles in algae are genetically determined, but the content of fatty acids also depends on the intensity of light and the salinity of water [39]. Ryckebosch et al. [40] reported that large amounts of EPA or DHA accumulate in the marine algae: Bacillariophyceae, Chlorophyceae, Chrysophyceae, Cryptophyceae, and Florideophyceae. Peltomaa et al. [39] suggested that freshwater algae may be more rich in ω-3 PUFAs than marine and brackish ones, and pointed freshwater diatom *Nitzschia* sp. as the most promising diatom strain for commercial EPA production. *Nitzschia laevis* contains 75.9% of EPA in total, and 37.4% of EPA is accumulated in the form of triacylglycerol, 22.6% in the form of monoacylglycerides, and 15.9% in the form of phosphatidylcholine [41]. In the *P. borealis* used in this study, the ω-3 PUFAs accounted for 28% of the total fatty acids and EPA accounted for 32% of all ω-3 PUFAs. 

Human EPA is obtained from the diet, mainly from fish oil. It is an essential fatty acid of important nutritional value and has the ability to alleviate the risk of many diseases. Thus, alternative sources to fish oil are needed to cover growing need for PUFA. The diatom *P. borealis* accumulates high levels (32%) of EPA and thus may be considered as a dietary supplement.

We found that diet supplementation with lyophilized diatom *P. borealis* solutions of different concentrations induced a dose-dependent decrease in cholesterol and triacylglycerol concentrations. Because of large variation in cholesterol content in the control group, this effect did not reach the level of significance in the liver (*p* = 0.07), but a consistent trend was evident. In the kidney, the cholesterol concentration in mice supplemented with the highest 5% diatom solution was decreased by ~30%. Consistently, triacylglycerol content was markedly reduced, both in the liver and kidney. We suggest that the decrease in cholesterol and triacylglycerol content was a result of diet supplementation with PUFAs, which are known for their protective effect [1]. *P. borealis* is an aerophytic diatom, which contains not only high concentration of EPA, but also several bioactive compounds, like pigments, fibers, and phytosterols, which all are beneficial for health [5,8]. Similar results were obtained by Sano et al. [42], who observed significant antilipidemic and antiatherosclerotic effects of diet supplementation with algae *Chlorella vulgaris* in rabbits fed a high-cholesterol diet for 10 weeks. Fallah et al. [43] demonstrated that *Chlorella* intake can also lower cholesterol levels in patients with hypercholesterolemia. These potential health benefits have been attributed to the specific components of *Chlorella*, like minerals, dietary fiber, proteins, and ω-3 PUFA. Cherng et al. [44] also showed that *Chlorella pyrenoidosa* has the ability to prevent dyslipidemia in rat and hamster models fed a high-fat diet containing 20% hydrogenated coconut oil (rich in saturated fatty acids). Several studies showed that animals fed high-fat diets are more likely to develop dyslipidemia and hyperglycemia [45], insulin resistance [46], hepatic steatosis [47], platelet aggregation [48], and oxidative stress [49]. Thus, a proper diet rich in PUFAs may ameliorate risk factors for these diseases.

Lee et al. [50] showed that in male Korean smokers, *Chlorella vulgaris* supplementation preserved plasma antioxidant nutrient status and improved erythrocyte antioxidant enzyme activities. Reactive oxygen species and free radicals may be harmful for lipids, proteins, and DNA and they may trigger a number of human diseases, like neurodegenerative and cardiovascular diseases [51]. In this study we predicted that diet supplementation with *P. borealis* would alleviate oxidative stress because PUFAs can decrease oxidative damage in vivo [1]. However, the concentration of malondialdehyde (MDA), a marker of oxidative stress, was the highest, both in the liver and kidney, in mice supplemented with the 5% diatom solution (Figure 2 and Figure 3). Malondialdehyde results largely from the peroxidation of PUFAs, and although it is less dangerous than previously thought, it is still toxic, immunogenic, and mutagenic [1]. However, MDA may come not only from lipid peroxidation, but also from food. Thus, it is plausible that the high content of MDA is not related to oxidative stress, but to the high EPA content in the dietary supplement. Indeed, the glutathione peroxidase (GPx) activity did not change or even decrease after diet supplementation (Figure 2). GPx enzymes catalyze the reduction of hydrogen peroxide protecting the organism from oxidative damage. Thus, if oxidative stress would be augmented, then GPx activity would increase. Incomprehensibly, GPx activity increased in the kidney of mice fed with the 3% diatom solution and it was paralleled by an increase in GSH content (Figure 3). Most probably, this is why the GSH concentration explained 23.7% of the variance in the kidney (Figure 3). These results could suggest an increase in oxidative stress in this group. However, glutathione reductase (GR) activity in the liver and kidney increased after diet supplementation, independent of the diatom solution (Figure 2). In the liver, GR activity explained 39.7% of the variance (Figure 2). This enzyme reconverted oxidized glutathione to its reduced form to maintain the proper oxidation-reduction potential, and protected the -SH groups of proteins against oxidation [22]. The ~2–2.5-fold increase in GR activity in all groups indicates that even with the lowest solution of *Pinnularia borealis,* antioxidant mechanisms are improved. The concentration of reduced glutathione (GSH) was decreased only in mice supplemented with the 5% diatom solution. It may suggest that lower concentrations are more beneficial. GSH is a potent antioxidant, which is also a cofactor for glutathione peroxidases and glutathione transferases [1]. Glutathione transferases are group of enzymes that initiate the detoxification of endogenous and exogenous cell-harmful compounds including reactive oxygen species. The highest activity of glutathione transferase (GST) in mice supplemented with the 5% diatom solution may also explain the lowest GSH content in these animals. Thus, we conclude that diet supplementation with *P. borealis* is beneficial for the redox system of GSH when applied in the solution up to 3%. The higher concentrations may have detrimental effects that cannot be compensated for by a decrease in cholesterol or triacylglycerol contents. However, dietary PUFAs can be incorporated into cell lipids to different extents, and as such may exert different effects depending on the cell structures or tissue [52]. This might partially explain the differences in antioxidant system parameters between the kidney and liver. In addition, fatty acids obtained from the diet may be desaturated or elongated, primarily in the endoplasmic reticulum [53]. We did not measure the EPA content in the liver and kidney, so we cannot say precisely what was the tissue EPA content, but we are convinced that the results were not accidental, because the lowest concentration of *P. borealis* was sufficient to improve antioxidant capacity.

In addition, there is growing evidence that different PUFAs used as dietary supplements are not equal, because PUFAs have not only pro-, but also antioxidant properties and, thus, may modulate endogenous antioxidant defense [52]. The effects of PUFAs are not simply related to the length of the carbon chain, or to the number and position of double bonds. Among five different PUFAs studied (arachidonic (20:4 ω-6), linoleic (18:2 ω-6), α-linolenic (18:3 ω-3), eicosapentaenoic (20:5 ω-3), and docosahexaenoic (22:6 ω-3)) only DHA increased antioxidant defense in a human hepatoma cell line culture, while arachidonic acid induced oxidative damage [51,54]. In other in vitro studies, the authors of [55] showed that exogenous EPA and DHA have the most potent antioxidant properties and they are much more effective in ROS scavenging than ω-6 PUFAs.

To the best of our knowledge, this is the first study on *P. borealis* as a food supplement. Algae can synthetize many compounds with potent health benefits [56]. Koyande et al. [45] showed that active compounds in algae have anticarcinogenic, antioxidative, and antihypertensive properties. One of the natural ways to eliminate the risks for neurodegenerative and cardiovascular diseases may be a diet rich in ω-3 PUFAs [38]. Some algae are classified as a food source in the GRAS category [57], and the diatom *P. borealis* has been recognized as a rich source of polyunsaturated fatty acids, such as EPA [58]. Nowadays, human EPA is obtained from the diet, mainly from fish oil. It is essential fatty acid of important nutritional value and ability to alleviate the risk of many diseases. Thus, alternative sources to fish oil are needed to cover growing need for PUFAs. The diatom *P. borealis* accumulates high levels (32%) of EPA and thus may be considered as dietary supplement.

## 5. Conclusions

The results of our study indicate that *Pinnularia borealis* can be used as a food supplement providing health benefits. When supplied in proper amount it can augment antioxidant defense and ameliorate risk factors for cardiovascular diseases. Algae are good alternative for fish oil, eliminating the problems of odor and instability of extraction products.

## Figures and Tables

**Figure 1 nutrients-13-01996-f001:**
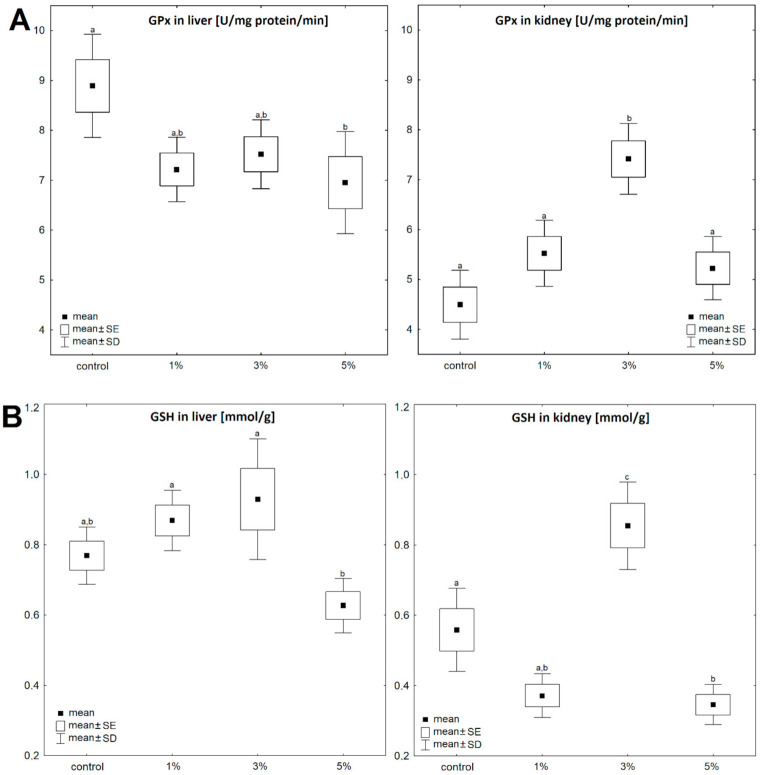
Effect of diet supplementation with lyophilized diatom solutions of different concentrations on glutathione peroxidase activity (GPx-U/mg protein/min), reduced glutathione (GSH-mmol/g tissue), glutathione reductase (GR-U/mg protein/min), glutathione transferase activity (GST-U/mg protein/min), malondialdehyde activity (MDA µmol/mg protein), cholesterol (µmol/g tissue), and triacylglycerols (µmol/g tissue) in mouse liver and kidney (*n* = 10). Different letters a,b,c) indicate statistically significant differences between groups (*p* < 0.05). (**A**)—glutathione peroxidase activity (GPx – U/mg protein/min), one-way ANOVA for liver: F (3, 36) = 3.80, *p* = 0.018; one-way ANOVA for kidney: F (3, 36) = 13.08, *p* = 6.2 × 10^−6^; (**B**)—reduced glutathione (GSH, mmol/g tissue), L: F (3, 36) = 5.44, *p* = 0.003; K: F (3, 36) = 23.12, *p* = 1.6 × 10^−8^; (**C**)—glutathione reductase (GR, U/mg protein/min), L: F (3, 36) = 29.34, *p* = 8.9 × 10^−10^; K: F (3, 36) = 9.85, *p* = 7.0 × 10^−5^; (**D**)—glutathione peroxidase activity (GST, U/mg protein/min), L: F (3, 36) = 11.62, *p* = 1.8 × 10^−5^; K: F (3, 36) = 15.59, *p* = 1.2 × 10^−6^; (**E**)—malondialdehyde activity (MDA, µmol/mg protein), L: F (3, 36) = 4.26, *p* = 0.011; K: F (3, 36) = 6.74, *p* = 0.001; (**F**)—cholesterol (µmol/g tissue), L: F (3, 36) = 2.57, *p* = 0.07; K: F (3, 36) = 14.90, *p* = 1.8 × 10^−6^; (**G**)—triacylglycerols (µmol/g tissue), L: F (3, 36) = 16.22, *p* = 7.8 × 10^−7^; K: F (3, 36) = 7.89, *p* = 0.0004.

**Figure 2 nutrients-13-01996-f002:**
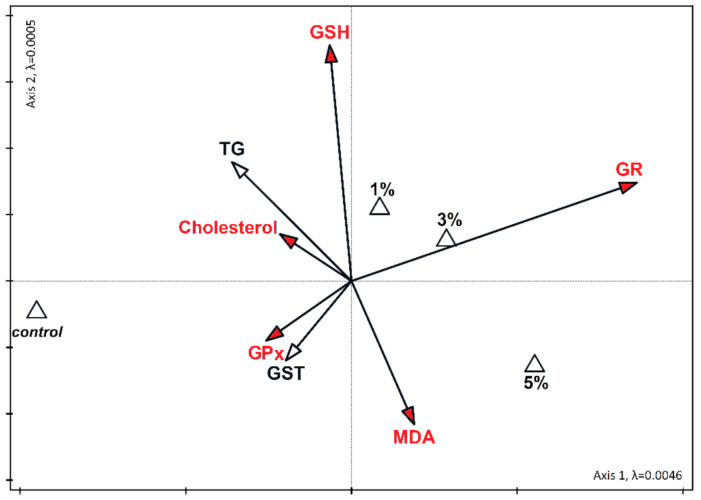
Canonical correspondence analysis (CCA) for the biochemical parameters in the liver of mice fed with different content of *Pinnularia borealis* (1%, 3%, and 5 %, and 0% for controls). Biochemical parameters are presented with vectors, and different algae concentrations in food are marked with triangles. The vectors of variables significantly differentiating the examined sets are marked in red. The eigenvalues canonical axes are marked in lambda (λ).

**Figure 3 nutrients-13-01996-f003:**
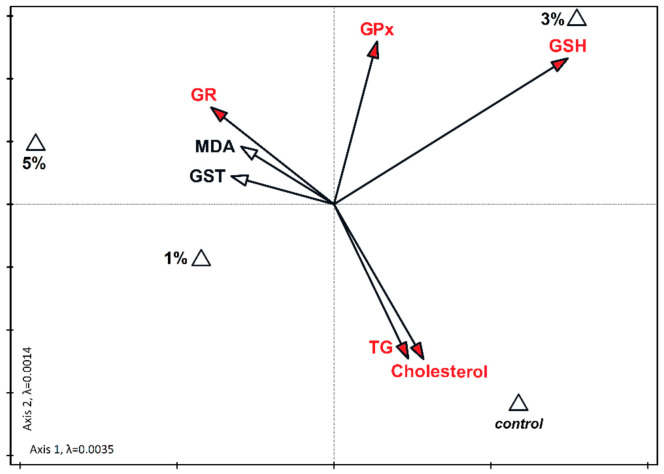
Canonical correspondence analysis (CCA) for the biochemical parameters in the kidneys of mice fed with different content of *Pinnularia borealis* (1%, 3%, and 5%, and 0% for controls). Biochemical parameters are presented with vectors, and different algae concentrations in food are marked with triangles. The vectors of variables significantly differentiating the examined sets are marked in red. The eigenvalues canonical axes are marked in lambda (λ).

**Table 1 nutrients-13-01996-t001:** Effect of diet supplementation with lyophilized diatom solutions of different concentrations on reduced glutathione (GSH-mmol/g tissue), glutathione peroxidase activity (GPx-U/mg protein/min), glutathione reductase (GR-U/mg protein/min), glutathione transferase (GST-U/mg protein/min), malondialdehyde activity (MDA µmol/mg protein), triacylglycerols (µmol/g tissue), and cholesterol (µmol/g tissue) in mouse liver (*n* = 10) and kidney (*n* = 10), expressed as means ± SD and as a percent of control values.

		GSH		GPx		GR		GST		MDA		Triacylglycerols		Cholesterol	
		x ± SD	%	x ± SD	%	x ± SD	%	x ± SD	%	x ± SD	%	x ± SD	%	x ± SD	%
**liver**	Control	0.77 ± 0.17	100	8.89 ± 2.06	100	1.92 ± 0.28	100	265.9 ± 49.9	100	2.12 ± 0.11	100	17.21 ± 4.06	100	7.15 ± 2.84	100
	Diatom 1%	0.87 ± 0.12	113	7.21 ± 1.68	81	4.89 ± 0.58	255	294.5 ± 43.1	111	2.27 ± 0.12	107	16.49 ± 2.41	96	6.67 ± 0.89	93
	Diatom 3%	0.93 ± 0.15	121	7.52 ± 0.76	85	4.87 ± 1.01	254	199.0 ± 38.2	75	2.68 ± 0.22	127	12.95 ± 3.71	75	6.28 ± 3.83	88
	Diatom 5%	0.63 ± 0.07	82	6.95 ± 1.01	78	5.56 ± 0.78	290	243.9 ± 33.4	92	2.88 ± 0.21	136	11.27 ± 5.08	66	5.50 ± 2.98	77
**kidney**	Control	0.56 ± 0.17	100	4.49 ± 1.37	100	0.34 ± 0.09	100	30.47 ± 6.07	100	3.16 ± 0.17	100	16.47 ± 6.22	100	6.13 ± 2.78	100
	Diatom 1%	0.37 ± 0.14	66	5.52 ± 1.12	123	0.63 ± 0.09	185	38.27 ± 5.82	126	3.57 ± 0.21	113	14.79 ± 5.16	90	5.49 ± 1.22	90
	Diatom 3%	0.86 ± 0.29	154	7.41 ± 1.54	165	0.51 ± 0.1	150	38.95 ± 6.13	128	3.77 ± 0.26	119	15.11 ± 1.91	92	5.55 ± 2.47	91
	Diatom 5%	0.33 ± 0.09	59	5.22 ± 1.51	116	0.61 ± 0.14	179	48.12 ± 5.41	158	4.61 ± 0.28	146	12.05 ± 5.37	73	4.23 ± 0.91	69

## Data Availability

The data presented in this study are available on request from the corresponding author.

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
