# Peer review of "Influence of Algae Supplementation on the Concentration of Glutathione and the Activity of Glutathione Enzymes in the Mice Liver and Kidney"

_nutrients, 2021, doi:10.3390/nu13061996_

Round 1

Reviewer 1 Report

Manuscript Number: nutrients-1214933

Authors: Grażyna Świderska-Kołacz, et al.

Title: Influence of Algae Supplementation on the Concentration of Glutathione

and the Activity of Glutathione Enzymes in the Mice Liver and Kidney: Case

Study of Oxidative Stress.

In this manuscript, authors studied whether diet supplementation with P. borealis may augment antioxidant defence and ameliorate risk factors for cardiovascular diseases using mice with lyophilized diatom solutions of different concentrations (1, 3 and 5 %) for 7 days. They found that cholesterol and triacylglycerol concentrations in liver and kidneys were the lowest in mice which were fed with the highest concentration of Pinnularia borealis, and the lowest concentration of Pinnularia borealis was sufficient to improve antioxidant capacity. They concluded that P. borealis may be used as a source for dietary supplements rich in EPA,

Although the aim of this study was interesting and important, there are some major and minor concerns, in which they did not demonstrate the direct relevance between omega3 fatty acids and the results.

Major concerns

  1. The experimental method is unclear. How were the mice fed with the solution of lyophilized diatom? , in food or in water?
  2. Actually, how much PUFA/day including EPA were administrated to the mice ?
  3. Did the dose-different solutions affect the food consumption, activity, and weight?
  4. Did the supplementation affect the distribution of PUFA in the liver and kidney?

Author Response

Answers to points raised by the Reviewers of manuscript ID: nutrients-1214933

"Influence of Algae Supplementation on the Concentration of Glutathione and the Activity of Glutathione Enzymes in the Mice Liver and Kidney” by Grazyna Świderska-Kołacz et al.

I would like to thank you very much for the remarks about our manuscript titled “Influence of Algae Supplementation on the Concentration of Glutathione and the Activity of Glutathione Enzymes in the Mice Liver and Kidney”. I am grateful to Reviewers for their comments. These comments allowed me to improve the quality of the manuscript. Below, I listed my answers to all questions and points raised by the Reviewers. All changes in revised manuscript are marked in green font).

Answers to the Reviewer’s comments:

Comment

Answer

Reviewer 1

The experimental method is unclear. How were the mice fed with the solution of lyophilized diatom?, in food or in water?

I the Material and Methods section we added the information that “Lyophilized P. borealis water solutions were given orally once a day in a volume of 50 µl.”

Actually, how much PUFA/day including EPA were administrated to the mice ?

The control group received a standard diet. The second group was fed the standard diet supplemented with 1% solution of lyophilized P. borealis (0.16 mg of EPA). The third group was fed the standard diet supplemented with 3% solution of lyophilized P. borealis (0.48 mg of EPA. The fourth group was fed the standard diet supplemented with 5% solution of lyophilized P. borealis (0.8 mg of EPA). Lyophilized P. borealis water solutions were given orally once a day in a volume of 50 µl.

Did the dose-different solutions affect the food consumption, activity, and weight?

The weight gain of the animals was monitored twice during the experiment and their food intake was also measured twice. However, we did not observe that the different solutions of diatoms affected the food consumption, activity and weight of mice. Our studies are preliminary, but we will continue the experiment (and we plan to extend it up to two weeks), and the result will be presented in the next manuscript.

Did the supplementation affect the distribution of PUFA in the liver and kidney?

Thank you for this comment. Unfortunately, we did not measure the distribution of PUFA in the liver and kidney. Our studies are preliminary, but we will continue it.

With many thanks for all comments.

Sincerely,

Joanna Czerwik-Marcinkowska

Reviewer 2 Report

The study by Świderska-Kołacz et al. examined the effect of dietary supplementation with diatom Pinnularia borealis on antioxidant defence in mouse Mus musculus. The study is carefully conducted. However, moderate English changes are required. Listed below there are some examples.

Page 2, line 48: “factors” instead of “factor”

Page 2, line 52: remove “the”

Page 2, line 67: “the” instead of “a”

Page 2, line 69: “consists” instead of “consist”

Author Response

Answers to points raised by the Reviewers of manuscript ID: nutrients-1214933

"Influence of Algae Supplementation on the Concentration of Glutathione and the Activity of Glutathione Enzymes in the Mice Liver and Kidney” by Grazyna Świderska-Kołacz et al.

I would like to thank you very much for the remarks about our manuscript titled “Influence of Algae Supplementation on the Concentration of Glutathione and the Activity of Glutathione Enzymes in the Mice Liver and Kidney”. I am grateful to Reviewers for their comments. These comments allowed me to improve the quality of the manuscript. Below, I listed my answers to all questions and points raised by the Reviewers. All changes in revised manuscript are marked in green font).

Answers to the Reviewer’s comments:

Reviewer 2

Page 2, line 48: “factors” instead of “factor”

Done.

Page 2, line 52: remove “the”

Done.

Page 2, line 67: “the” instead of “a”

Done.

Page 2, line 69: “consists” instead of “consist”

Done.

Reviewer 3 Report

In the manuscript “Influence of Algae Supplementation on the Concentration of Glutathione and the Activity of Glutathione Enzymes in the Mice Liver and Kidney: Case Study of Oxidative Stress” Authors measured some biological responses related to Glutathione metabolisms, lipid peroxidation, cholesterol and triacylglycerols in two different tissues of mice exposed to 3 different concentration of diatom Pinnularia borealis.

The principal gaps of this manuscript are the results presentation and section.

Please remove tables 1, 2 and 3 and replace with figures from supplementary material (in this way remove completely Supplementary materials file). Please move Figures 1 and 2 (CCA) after histogram figures of analyzed parameters. Please rewrite newly the results section in scientific way, actually this section appeared unclear. Please the phrase “is also not accidental” what it means? It is a scientific terminology? I don’t believe.  Please present the results of Canonical Correspondence Analysis (CCA).

Please add protein content protocols in material and methods section.

Author Response

Answers to points raised by the Reviewers of manuscript ID: nutrients-1214933

"Influence of Algae Supplementation on the Concentration of Glutathione and the Activity of Glutathione Enzymes in the Mice Liver and Kidney” by Grazyna Świderska-Kołacz et al.

I would like to thank you very much for the remarks about our manuscript titled “Influence of Algae Supplementation on the Concentration of Glutathione and the Activity of Glutathione Enzymes in the Mice Liver and Kidney”. I am grateful to Reviewers for their comments. These comments allowed me to improve the quality of the manuscript. Below, I listed my answers to all questions and points raised by the Reviewers. All changes in revised manuscript are marked in green font).

Answers to the Reviewer’s comments:

Reviewer 3

Please remove tables 1, 2 and 3 and replace with figures from supplementary material (in this way remove completely Supplementary materials file).

I apologize for the confusion with tables and figures from supplementary materials. I deleted all tables from the manuscript. I corrected the numeration of all figures in manuscript.

Please move Figures 1 and 2 (CCA) after histogram figures of analyzed parameters.

Done.

Please rewrite newly the results section in scientific way, actually this section appeared unclear.

I corrected the results section according to suggestion.

Please present the results of Canonical Correspondence Analysis (CCA).

Thank you for your comments. I made  changes to Figures 2 and 3.

Please add protein content protocols in material and methods section.

Done.

Please the phrase “is also not accidental” what it means? It is a scientific terminology? I don’t believe. 

This phrase was deleted. We added: “The increase is the levels of GSH and GR at low concentrations of diatoms in the feed is also significant.”

Reviewer 4 Report

In this paper, the Authors investigated the effect on antioxidant and oxidant parameters, as well lipid content, in liver and kidney of Swiss mice fed with different concentration of lyophilized diatom solution.

In liver, algae supplementation reduced triacylglycerols, Gpx and  GST activity and increased GR activity and MDA content. On the contrary, in kidney algae supplementation increased antioxidant and detoxyfing enzymes as well MDA and reduced lipid content. In discussion section, the Authors attributed these effects particularly to EPA, which algae are particularly rich. The study is quit elegant, the methodologies used are pertinent and well described and gives new insight on the biological effect of algae to be used as functional ingredient or supplement.

Despite these considerations, major issue raised.

Major considerations.

The modulatory role of PUFA on endogenous antioxidant defense is intriguing although not entirely new (10.1016/j.phrs.2008.05.002; 10.1080/09637486.2016.1201790; 10.1016/j.plefa.2011.07.005). The Authors should consider including these references in discussion section.

As algae contain bioactive compounds which may modulate antioxidant defenses (selenium, carotenoids…) a fatty acid composition of liver and kidney tissue is mandatory to verify the real fatty acid incorporation and to attribute these effects to PUFA. In fact, the different degree of fatty acid incorporation may at least in part explain the different effects between liver and kidney. Moreover the Authors are inclined to attribute the biological effects to EPA which are most present fatty acid in algae without considering that at cell level fatty acid may  elongated and desaturated and thus the cellular fatty acid composition not necessarily reflect those present in the supplement.

Can the levels of algae supplementation for mice be achieved within a normal human diet? This point is fairly important to have technological, economic and health effects in the population.

Minor considerations.

Introduction section contains many pleonastic sentences. Please rewrite in a more concise form.

Title: Include the term “oxidative stress” can be misleading. First, an exogenous oxidative stress was not applied. Second, it seems that algae supplementation may cause an oxidative stress. I suggest removing it from the title.

The Authors should provide the complete fatty acid composition of algae (at least as supplementary), not only the content of EPA and PUFA as reported in line 116 and 117.

Figures (as supplementary) and tables are at the same time present to describe the same results. The Authors should choose only one. If the tables will be chosen, the Authors should include SD in percentage value and statistical symbols.

Line 243-244. It is not clear the meaning of the sentence.

Author Response

Answers to points raised by the Reviewers of manuscript ID: nutrients-1214933

"Influence of Algae Supplementation on the Concentration of Glutathione and the Activity of Glutathione Enzymes in the Mice Liver and Kidney” by Grazyna Świderska-Kołacz et al.

I would like to thank you very much for the remarks about our manuscript titled “Influence of Algae Supplementation on the Concentration of Glutathione and the Activity of Glutathione Enzymes in the Mice Liver and Kidney”. I am grateful to Reviewers for their comments. These comments allowed me to improve the quality of the manuscript. Below, I listed my answers to all questions and points raised by the Reviewers. All changes in revised manuscript are marked in green font).

Answers to the Reviewer’s comments:

Reviewer 4

The modulatory role of PUFA on endogenous antioxidant defense is intriguing although not entirely new (10.1016/j.phrs.2008.05.002; 10.1080/09637486.2016.1201790; 10.1016/j.plefa.2011.07.005). The Authors should consider including these references in discussion section.

Thank you for comments. I added important information about modulatory role of PUFA on endogenous antioxidant system in Discussion section using suggested references.

52. Di Nunzio, M.; Valli, V.; Bordoni, A. Pro- and anti-oxidant effects of polyunsaturated fatty acid supplementation in HEPG2 cells. Prostaglandins, Leukotrienes and Essential Fatty Acids 2011, 85, 121–127.

53. Bond, L.M.; Miyazaki, M.; O’Neill, L.M.; Ding, F.; Ntambi, J.M. Fatty Acid Desaturation and Elongation in Mammals. In Biochemistry of Lipids, Lipoproteins and Membranes, 6th ed.; N.D., Ridgway, R.S., McLeod; Elsevier 2016, 185–208.

54. Di Nunzio, M.; Valli, V.; Bordoni, A. PUFA and oxidative stress. Differential modulation of cell response by DHA. International Journal of Food Sciences and Nutrition 2016, 67, 7, 834–843.

55. Richard, D.; Kefi, K.; Barbe, U.; Bausero, P.; Visioli, F. Polyunsaturated fatty acids as antioxidants. Pharmacological Research 2008, 57, 451–455.

As algae contain bioactive compounds which may modulate antioxidant defenses (selenium, carotenoids…) a fatty acid composition of liver and kidney tissue is mandatory to verify the real fatty acid incorporation and to attribute these effects to PUFA. In fact, the different degree of fatty acid incorporation may at least in part explain the different effects between liver and kidney. Moreover the Authors are inclined to attribute the biological effects to EPA which are most present fatty acid in algae without considering that at cell level fatty acid may  elongated and desaturated and thus the cellular fatty acid composition not necessarily reflect those present in the supplement.

Thank you very much for comment. I added more information in Materials and Methods and in Discussion sections.

“However, dietary PUFA can be incorporated into cell lipids to different extent and as such may exert different effects depending on cell structures or tissue [51]. This might partially explain the differences in antioxidant system parameters between kidney and liver. Also, fatty acids obtained from the diet may be desaturated or elongated, primarily in the endoplasmic reticulum [52]. We did not measure EPA content in liver and kidney, so we cannot say precisely what was the tissue EPA content, but we are convinced that the results were not accidental because the lowest concentration of P. borealis was sufficient to improve antioxidant capacity.

Also, there is growing evidence that different PUFA used as dietary supplements are not equal because PUFA have not only a pro- but also anti-oxidant properties and thus may modulate endogenous antioxidant defense [51]. The effects of PUFA are not simply related to the length of carbon chain, or to the number and position of double bonds. Among five different PUFA studied (arachidonic (20:4 ω-6), linoleic (18:2 ω-6), α-linoleic (18:3 ω-3), eicosapentaenoic (20:5 ω-3), docosahexaenoic (22:6 ω-3) only DHA increased antioxidant defence in human hepatoma cell line culture, while arachidonic acid induced oxidative damage [51,53]. In other in vitro studies, [54] and co-workers showed that exogenous EPA and DHA have the most potent antioxidant properties and they are much more effective in ROS scavenging that ω-6 PUFAs.”        

Can the levels of algae supplementation for mice be achieved within a normal human diet? This point is fairly important to have technological, economic and health effects in the population.

Thank you very much for comment. The  demand for algae foods is growing and algae are increasingly being consumed for the functional benefits beyond the traditional considerations of nutrition and health. There is evidence for the health benefits of algal supplementation but there is lack of quantifying these benefits, as well as possible adverse effects. There is a limited understanding of nutritional composition across algal species. The diatoms (P. borealis) accumulate high levels (32%) of EPA and thus may be considered as dietary supplement. Our preliminary calculation show that 3.5 ml of algae supplementation can be equivalent to a level of 1 kg per human weight.

Introduction section contains many pleonastic sentences. Please rewrite in a more concise form.

I modified this paragraph. I hope that is much better now.

Title: Include the term “oxidative stress” can be misleading. First, an exogenous oxidative stress was not applied. Second, it seems that algae supplementation may cause an oxidative stress. I suggest removing it from the title.

I apologize for the confusion with term “oxidative stress” using in the title. We modified title according to suggestion.

The Authors should provide the complete fatty acid composition of algae (at least as supplementary), not only the content of EPA and PUFA as reported in line 116 and 117.

Thank you for your comments. I added this information in the Materials and Methods section.

Figures (as supplementary) and tables are at the same time present to describe the same results. The Authors should choose only one. If the tables will be chosen, the Authors should include SD in percentage value and statistical symbols.

Thank you for your comments. I deleted the all Tables and modified or added Figures 1-3.

Line 243-244. It is not clear the meaning of the sentence.

This sentence was deleted.

With many thanks for all comments.

Sincerely,

Joanna Czerwik-Marcinkowska

Round 2

Reviewer 1 Report

I recommended that the data, as suggested previously, should be included to estimate the results.

Author Response

I recommended that the data, as suggested previously, should be included to estimate the results.

I added data on body mass and food consumption at the beginning of the Result section. I hope that I understood correctly this comment.

In Result section: “Before experiment mice weighed 20.5g±0.5g. After one week of supplementation with different diatom solutions, mice did not differ in body mass [t = -1.75, p = 0.088]. Food consumption also did not differ between groups and mice ate on average 2.5 +/- 0.4 g of food per day.

In Materials and Methods section (3.1. Statistical Analysis): “Biochemical parameters were compared using analysis of variance ANOVA followed by a Tukey post hoc test. Changes in body mass and food consumption were compared using two-sample pair t-test.”

Reviewer 3 Report

The Authors accepted all of the reviewer's revisions; now the manuscript can be published as it is.

Author Response

Answers to the Reviewer’s comments:

Reviewer 3

The Authors accepted all of the reviewer's revisions; now the manuscript can be published as it is.

With many thanks for all comments.

Sincerely,

Joanna Czerwik-Marcinkowska

Reviewer 4 Report

The manuscript is significantly improved.

Anyway some minor corrections are still required.

Line 121. Include the “stenographic” nomenclature for DHA and remove EPA at the end (just included)

Line 227. Replace the comma in “1,3” with a dot.

Line 210-212. The sentence is not clear. Is Lowry’s method sensible to peptide bond, aromatic amino acid (phenylalanine is missing) and also to cysteine?

Figure 1. Figure legends is randomly present in only some graphs. Please uniform.

Line 350. Comma after “diet” is not necessary

Line 361. Comma after “plausible” is not necessary.

Line 395. It is alpha-linolenic and not alpha-linoleic

Line 401. Koyande does not correspond to ref 44

Author Response

Line 121. Include the “stenographic” nomenclature for DHA and remove EPA at the end (just included).

Thank you for this comment. We added “docosahexaenoic acid” for DHA and I deleted “EPA” at the end sentence.

Line 227. Replace the comma in “1,3” with a dot.

Done.

Line 210-212. The sentence is not clear. Is Lowry’s method sensible to peptide bond, aromatic amino acid (phenylalanine is missing) and also to cysteine.

I apologize for the confusion with Lowry’s method. We added more information: “The Lowry protein assay is based on the reactions of copper ions with the peptide bonds (with Folin-Ciocalteu reagent i.e. a mixture of phosphotungstic acid and phosphomolybdic acid) under alkaline conditions (the Biuret test) with the oxidation of aromatic protein residues (mainly tryptophan and tyrosine). Cysteine is also reactive to the reagent.

Figure 1. Figure legends is randomly present in only some graphs. Please uniform.

Done.

Line 350. Comma after “diet” is not necessary.

Done.

Line 361. Comma after “plausible” is not necessary.

Done.

Line 395. It is alpha-linolenic and not alpha-linoleic

Done. I apologise for the mistake.

Line 401. Koyande does not correspond to ref 44

The correct number is 45.

With many thanks for all comments.

Sincerely,

Joanna Czerwik-Marcinkowska